# Mitochondrial dysfunction in rheumatoid arthritis: A comprehensive analysis by integrating gene expression, protein-protein interactions and gene ontology data

Venugopal Panga[1,2], Ashwin Adrian Kallor[1], Arunima Nair[1], Shilpa Harshan[1,2], Srivatsan Raghunathan[1] *

1 Institute of Bioinformatics and Applied Biotechnology (IBAB), Bengaluru, Karnataka, India, 2 Manipal Academy of Higher Education, Manipal, Karnataka, India

* srivatsan@ibab.ac.in

**Data Availability Statement:** All relevant data are within the manuscript and its Supporting Information files.

## Abstract

Several studies have reported mitochondrial dysfunction in rheumatoid arthritis (RA). Many nuclear DNA (nDNA) encoded proteins translocate to mitochondria, but their participation in the dysfunction of this cell organelle during RA is quite unclear. In this study, we have carried out an integrative analysis of gene expression, protein-protein interactions (PPI) and gene ontology data. The analysis has identified potential implications of the nDNA encoded proteins in RA mitochondrial dysfunction. Firstly, by analysing six synovial microarray datasets of RA patients and healthy controls obtained from the gene expression omnibus (GEO) database, we found differentially expressed nDNA genes that encode mitochondrial proteins. We uncovered some of the roles of these genes in RA mitochondrial dysfunction using literature search and gene ontology analysis. Secondly, by employing gene co-expression from microarrays and collating reliable PPI from seven databases, we created the first mitochondrial PPI network that is specific to the RA synovial joint tissue. Further, we identified hubs of this network, and moreover, by integrating gene expression and network analysis, we found differentially expressed neighbours of the hub proteins.

The results demonstrate that nDNA encoded proteins are (i) crucial for the elevation of mitochondrial reactive oxygen species (ROS) and (ii) involved in membrane potential, transport processes, metabolism and intrinsic apoptosis during RA. Additionally, we proposed a model relating to mitochondrial dysfunction and inflammation in the disease. Our analysis presents a novel perspective on the roles of nDNA encoded proteins in mitochondrial dysfunction, especially in apoptosis, oxidative stress-related processes and their relation to inflammation in RA. These findings provide a plethora of information for further research.

**Funding:** We thank the department of Information Technology, Biotechnology and Science & Technology (IT, BT and S&T), Government of Karnataka, India for infrastructure support. VP received fellowships from the Institute of Bioinformatics and Applied Biotechnology (IBAB) as well as from the Council of Scientific and Industrial Research (CSIR), Government of India (GoI) (File No. 09/1086(0001)/2012-EMR-1), URL: http://www.csirhrdg.res.in/. SR is a faculty at IBAB. This project was partially supported by a grant from the Department of Biotechnology, GoI (BTPR12422/MED/31/287/2014, URL: http://www.dbtindia.nic.in/. The funders had no role in study design, data collection and analysis, decision to publish, or preparation of the manuscript.

**Competing interests:** The authors have declared that no competing interests exist.

## Introduction

Mitochondrial dysfunction prevails among numerous diseases, including RA, Sjøgren's syndrome, neurodegenerative diseases, diabetes, cancer and obesity [1–5]. Genomic technologies and computational approaches played a vital role in our understanding of mitochondrial dysfunction in several diseases like Leigh syndrome, cardiovascular diseases, obesity and infantile-onset mitochondrial encephalopathy [6–12]. These approaches have also discerned the mechanics of calcium uniporter in mitochondrial biology and associated diseases [7, 13–15]. Further, investigations into metabolic profiling and whole-exome sequencing data point to metabolic abnormalities concerned with mitochondria and biallelic mutations leading to instability in mitoribosomal subunits in Leigh syndrome [6, 8].

Mitochondria, which are membrane-bound cell organelles, are the primary generators of adenosine triphosphate (ATP). The respiratory chain complexes, which are part of the mitochondrial oxidative phosphorylation (OxPhos), are necessary for the production of ATP. The genome of this organelle has 13 protein-coding genes, which are associated with the OxPhos pathway. It is understood that 1158 nDNA encoded proteins get translocated to this cell organelle [16], and some of them are crucial for the OxPhos pathway. However, the functional roles of many of these proteins in RA mitochondrial dysfunction are uncertain, creating a serious lacuna in our understanding of this disease. An integrative analysis of these proteins using gene expression, PPI, gene ontology and network theory offers an excellent opportunity for deducing some of their roles.

About 1% of the world's population is affected by RA [17]. It is a chronic inflammatory disease that usually affects the small synovial joints of the hands and feet. The disease synovium gets inflamed (a condition called synovitis) and invades articular cartilage and bone, forming a layer of granulation tissue called pannus. Further, synovitis causes irreversible damage to the synovium in joints [18]. Moreover, the cells of the RA synovium (synoviocytes) secrete inflammatory cytokines and articular cartilage-degrading enzymes, such as matrix metalloproteinases (MMPs), which further aggravate the disease.

The composition of cell types in a healthy synovium is different to that of RA. The healthy synovium primarily contains two cell types, macrophage-like synoviocytes (MLS) and fibroblast-like synoviocytes (FLS) [19]. Other cell types such as leucocytes can be seen in small numbers [19]. In contrast, the RA synovium is expanded and forms pannus and contains resident MLS and FLS as well as heavily infiltrated leucocytes [20–21].

The pannus in RA, like a tumour, increases demand for energy (ATP) in the synovium. Additionally, the dysregulated synovial microvasculature results in a poor supply of oxygen to the tissue, causing hypoxia. Both the increased energy demand on mitochondrial electron transport and hypoxia could lead to an enhanced production of ROS, creating oxidative stress in synoviocytes [1]. Further, an inverse correlation between synovitis and the partial pressure of oxygen in the synovium testifies to the role of hypoxia in arthritis [22]. Moreover, hypoxia might induce proinflammatory pathways, through hypoxia-inducible factor-1α (HIF-1α), nuclear factor κB (NF-κB), Janus kinase-signal transducer and activator of transcription (JAK-STAT), activator protein 1 (AP-1) and Notch. Most notably, anti-tumour necrosis factor therapy has significantly decreased synovial hypoxia in vivo, indicating that it is a crucial event in arthritis [1, 22–25]. This elucidates that hypoxia and ROS are relevant to RA mitochondrial dysfunction.

Superoxide anion ($O_2 \cdot^-$), hydrogen peroxide ($H_2O_2$) and hydroxyl radical ($\cdot OH$) are collectively called ROS [26–27]. The components of ROS can damage DNA, proteins, lipids and many other molecules. Synovial fluid (SF) and plasma samples as well as blood lymphocytes and polymorphonuclear leucocytes from RA patients have significantly higher mitochondrial

DNA (mtDNA) and oxidatively damaged DNA adduct, 8-hydroxyl-2′-deoxyguanosine (8-oxodG), than non-arthritic samples. Further, both the mtDNA and 8-oxodG levels in SF correlate with the presence of rheumatoid factor in RA patients [28–29]. This underlines the existence of ROS-mediated damage of mitochondria in this disease. Other oxidative stress markers, such as protein carbonyls are significantly higher in the serum of RA patients compared to healthy controls. Treatment of these patients with infliximab resulted in a significant decrease of the carbonyls [30]. Iron, a catalyst for the formation of ·OH from $H_2O_2$ via the Fenton's reaction, is present in the diseased synovium [31]. Oxidised low-density lipoproteins and lipid peroxidation as well as the latter's correlation with the concentration of Iron ions were observed in SF of RA patients [32–33]. Furthermore, hyaluronate-derived small oligosaccharides are present in the inflamed disease joints, revealing the activity of ROS [34]. The ROS-mediated damage of mitochondria might also result in angiogenesis and cartilage destruction, the latter of which ensues through the up-regulation of MMPs [1, 25, 35–36]. Besides, there is an inverse association between the dietary intake of antioxidants and the prevalence of RA as well as the levels of antioxidants and the disease inflammation [37–42]. Moreover, an element of ROS, $O_2\cdot^-$ reacts with nitric oxide (NO) to form peroxynitrite ($ONOO^-$), which is a component of the reactive nitrogen species (RNS). This reactive species plays a role in the NF-κB-mediated production of inflammatory mediators, such as tumour necrosis factor (TNF), interleukin-1 beta (IL-1β) and inducible nitric oxide synthase (iNOS) [43].

To summarise, the pannus increases ATP demand and the dysregulated microvasculature creates hypoxia. Both the conditions can generate ROS in RA synovial mitochondria and the immediate targets of these free radicals are mtDNA, proteins and lipids. Supporting this phenomenon in RA, elevated levels of the damaged mtDNA, proteins and lipids were observed in SF, plasma and leucocytes of the patients. Additionally, both the hypoxia and ROS are known to induce pro-inflammatory HIF-1α, NF-κB, JAK-STAT, AP-1 and Notch pathways. As stated earlier, 1158 nDNA encoded proteins get translocated to mitochondria and several of them could be involved in the pathways concerned with the generation of ROS. So, it is of great pathophysiological relevance to elucidate the roles of these proteins in mitochondrial dysfunction and their connection to ROS-mediated damage, hypoxia, ATP synthesis and inflammation in RA.

In RA, apoptosis is required to control synovial hyperplasia. Apoptosis can occur by two different pathways, the extrinsic and the intrinsic, of which the latter could be initiated in mitochondria in response to oxidative stress. Both the pathways culminate in the activation of a cascade of proteases, called caspases. It has been shown that the extrinsic pathway is inactive in RA FLS [44]. Fas, which is a pro-apoptotic molecule and known to be involved in the extrinsic pathway, has been found to induce inflammation rather than apoptosis in RA FLS. However, this process depends on caspase-8 (CASP8) activity and FLICE-like inhibitory protein (FLIP) expression [45]. Therefore, studying the intrinsic pathway might give clues on the regulation of synovial hyperplasia. Hence it is important to understand the roles of the nDNA encoded proteins that could be implicated in this pathway.

In the current study, we followed an integrated approach that uses microarray data, PPI, gene ontology and network analysis. Six microarray gene expression datasets related to RA and healthy synovium were obtained from GEO, and they were analysed to discover differentially expressed genes (DEGs) encoding mitochondrial proteins. Further, a mitochondrion-specific PPI network has been created based on the information from seven publicly available databases. We also performed gene ontology analysis (GO) using the Search Tool for the Retrieval of Interacting Genes (STRING) database for identifying significantly enriched biological processes (BP), molecular functions (MF) and cellular components (CC). In addition to a discussion on the roles of nDNA encoded proteins in RA mitochondrial dysfunction based

on available information in the literature, a model for the relation between mitochondrial dysfunction and the disease inflammation has also been framed.

## Methods

### Data collection

The mRNA expression datasets (GSE77298, GSE7307, GSE12021, GSE55235 and GSE55457) were retrieved from GEO, which is a public National Center for Biotechnology Information (NCBI) database. Table 1 gives a detailed account of the mRNA expression datasets. All the datasets were downloaded in raw data file format for analysis.

### Construction of mitochondrial PPI network in RA synovium

The mitochondrial PPI network in RA synovium was created by pooling the experimentally determined interactions in human cells. They were obtained from seven publicly available resources, namely the biological general repository for interaction datasets (BioGRID), IntAct, the molecular interaction (MINT), STRING, the human protein reference database (HPRD), the database of interacting proteins (DIP) and CRG [46–52]. Among them, the first four databases have confidence scores for each interaction. The higher the score the more is the confidence for the interaction to occur. For the current study, from each of these four, we have got more reliable interactions by putting a cut-off to the confidence scores. The cut-off was decided in such a way that the interactions having a confidence score more than the median of the score distributions were selected. From DIP, the interactions with the core quality status were considered. From HPRD and CRG, which do not have confidence scores, only those interactions which have at least two publication evidences were considered. Collectively, a total of 387,242 interactions were obtained from all the seven resources. Then, to create the mitochondrial PPI network, only the interactions of those proteins that get localised to mitochondria were chosen using MitoCarta [16], which is a compendium of 1158 nDNA genes that encode mitochondrial proteins.

Furthermore, to make this interactome specific to the synovial tissue, we measured the co-expression of the interacting partners of these interactions using the gene expression data from six microarray datasets. Table 1 gives detailed information about these datasets. For each dataset, the raw intensities were normalised using the RMA algorithm. For the interacting partners of each interaction, we computed the Pearson correlation coefficient of the normalised expression values across all disease samples. Only those interactions with a Pearson correlation coefficient > 0.7 between the partners, in at least one microarray dataset, were considered co-expressed in the synovial tissues. The resulting interactions were used to create the undirected mitochondrial PPI network, using the 'igraph' package in R. The hubs of this network were identified using the same package in R.

**Table 1. Details of microarray datasets used in this study.**

| S.No. | GEO Accession | PubMed ID | Microarray Platform | Probe Number | Number of Samples | |
|---|---|---|---|---|---|---|
| | | | | | RA | Control |
| 1 | GSE77298 | 26711533 | Affymetrix Human Genome U133 Plus 2.0 Array | 54675 | 16 | 7 |
| 2 | GSE7307 | - | Affymetrix Human Genome U133 Plus 2.0 Array | 54675 | 5 | 9 |
| 3 | GSE12021 | 18721452 | Affymetrix Human Genome U133A Array | 22283 | 12 | 9 |
| 4 | GSE12021 | 18721452 | Affymetrix Human Genome U133B Array | 22645 | 12 | 4 |
| 5 | GSE55457 | 24690414 | Affymetrix Human Genome U133A Array | 22283 | 13 | 10 |
| 6 | GSE55235 | 24690414 | Affymetrix Human Genome U133A Array | 22283 | 10 | 10 |

## Differential expression analysis of microarray data

The microarray experiments, considered in this study, were carried out on RA and normal synovial tissues by other workers (Table 1). The RA samples used in these studies were obtained by tissue excision upon joint replacement/synovectomy surgery from RA patients. Similarly, the control samples were obtained from either postmortem or traumatic joint injury cases. In four of the six datasets (GSE12021 (HGU133A), GSE12021 (HGU133B), GSE55235 and GSE55457), for which the information on duration and severity of the disease is available, the duration of the disease in the patients was reported to be a mean of at least 12 years. The number of American rheumatism association (ARA) (now, American college of Rheumatology) criteria for RA was reported to be a mean of at least five [53–54]. The patients, who participated in five of the six studies, are from the Netherlands and Germany. For one study (GSE7307), the demography of patients is not available.

We re-analysed all the datasets using the R/Bioconductor statistical package. The intensities were normalised using two algorithms, MAS5 and RMA, separately. The differential expression of the genes between RA and control groups was computed using the two sample independent t-test. A p-value < 0.05 and a fold-change of > 1.5 in the up or down direction were taken as the cut-off values for differential expression. Further, the following conditions were imposed for deciding a differentially expressed gene across the datasets:

1. For one dataset, if the gene is selected by both the normalisation methods in the same direction (up or down)

2. For multiple datasets, if the gene is selected by both the normalisation methods in at least one dataset or by complementary normalisation methods in at least two datasets.

3. If the gene is up-regulated in at least one dataset and not down-regulated in any of the remaining, we call it a consistently up-regulated gene. A similar criterion was applied for a down-regulated gene. On the other hand, if a gene shows up-regulation in some and down-regulation in the other datasets, we call it a mixed-regulated gene.

Since we are particularly interested in nuclear genes that encode mitochondrial proteins, and in order to maximise the DEGs of mitochondrial proteins, we did not correct the p-value. However, for most of the analyses performed, the genes that were selected in at least two or three datasets were considered. Further, the DEGs were used for integrative analysis and hence the false positives might be reduced.

## Gene ontology and pathway enrichment analysis

The GO and pathway enrichment analyses were carried out using the STRING database. These analyses identify enriched GO terms and KEGG (Kyoto Encyclopedia of Genes and Genomes) pathways for a given list of genes by employing a hypergeometric test that was discussed elsewhere [55–56]. A false discovery rate (FDR) < 0.01 was considered as the cut-off for the significantly enriched GO terms and pathways.

## Results

### Creation of mitochondrial PPI network in the RA synovium

High-confident PPI from seven public resources, namely BioGRID, IntAct, MINT, STRING, HPRD, DIP and CRG, were used to construct this network [46–52]. The first four databases provide a confidence score, which is a measure of reliability, for each interaction. From each of these databases, only the interactions with the scores above the median of the confidence score

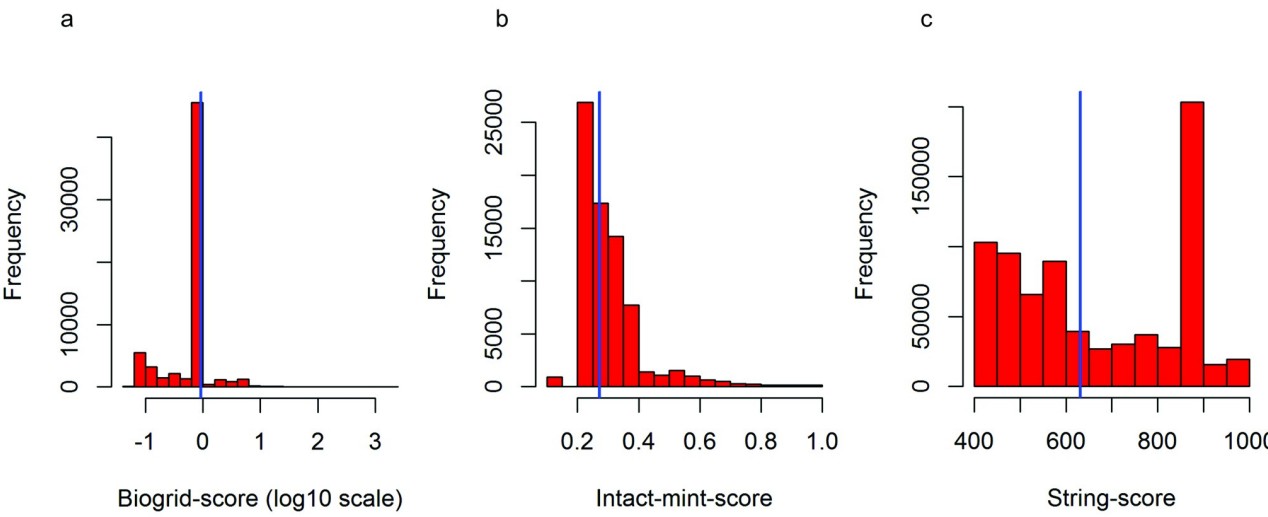

**Fig 1. Confidence score distributions of PPI in (a) Biogrid, (b) Intact and Mint, and (c) String.** The blue vertical line in all of them corresponds to the median of the distributions. The interactions which have a confidence score above the median were considered for the current study.

distributions were extracted ([Fig 1]). From DIP, only those interactions which have core quality status (a reliability parameter specific to DIP) were extracted. From HPRD and CRG, the interactions with at least two publication evidences were selected. This resulted in 387,242 reliable interactions which constitute ~40% of the total interactions in these databases. Then, we applied a filter requiring both the interacting partners to be nDNA encoded mitochondrial proteins, the list of which could be found in MitoCarta [16]. This has returned an interactome with 7023 interactions, representing 926 of the 1158 MitoCarta genes (79.96%). In order to make the interactome specific to the RA synovium, we computed the co-expressions between the interacting partners using RA synovial microarray data ([Table 1]). For each pair of interacting proteins, their expression levels across the RA disease samples in a given microarray dataset were used to compute the Pearson's correlation coefficient ($\rho$) as a measure of co-expression. The interacting partners with a $\rho > 0.7$ between their gene expression values in at least one of the six microarray datasets were considered to be co-expressed in the synovium. This selection criterion, which was chosen to maximise the number of PPI with the co-expressed interacting partners, resulted in an interactome with 2708 interactions and 665 genes, representing 57.42% of MitoCarta genes. In this interactome, on an average, each protein is connected to four other proteins. None of the interacting partners were co-expressed in all the six datasets. Of the 2708 interactions, 13 have the co-expressed interacting partners in five microarray datasets; 65 in four; 184 in three; 618 in two and 1828 in one datasets. With these interactions, we have created the mitochondrial PPI network using the 'igraph' package in R. In order to identify the localisation of the network proteins in mitochondria, a GO analysis for the cellular component (CC) term was performed using the STRING database [55–56]. It was observed that the majority of the network proteins get translocated to the mitochondrial inner membrane and matrix ([S1 Fig]).

## Differential expression of nuclear genes encoding mitochondrial proteins

Differential expression analysis between RA and healthy human synovial tissue samples was carried out using the six microarray datasets obtained from GEO ([Table 1]). The requirement for differential expression in a dataset was set to be a fold-change > 1.5 (up or down

regulation) and a p-value $< 0.05$. Of the 665 PPI network genes, 131 were found to be differentially expressed in at least one RA synovial microarray dataset (S1 Table). The criterion of a gene having a differential expression in at least one of the six datasets was decided so as to maximise the selection of mitochondrial DEGs. The whole network indicating the up and down DEGs, which can be visualised using the Cytoscape tool, is in S1 File. The 131 genes include 46 consistently up-, 73 consistently down- and 12 mixed-regulated genes (for methodological details, see 'Methods' section). Another 77 of the MitoCarta members, which are not part of the network, were also differentially expressed in at least one dataset (S2 Table). Thus, it makes a total of 208 mitochondrial DEGs (83 up-, 111 down- and 14 mixed-regulated). Their differential expression across the six studies is as follows; only one of the 208 genes was a DEG in all the six studies; three (1.44%) genes in five studies; four (1.92%) in four; 15 (7.21%) in three; 60 (28.84%) in two and 125 (60.09%) in one study.

Mitochondrial DEGs that were found in at least three datasets are in Table 2 (13 up-, 6 down- and 4 mixed-regulated). Among them, the following up-regulated genes are highlighted in respect of their functions in mitochondria. Acyl-CoA thioesterase 7 (ACOT7) is an enzyme that hydrolyses long-chain fatty acids such as palmitoyl-CoA. Kynurenine 3-monooxygenase (KMO) is an enzyme that catalyses the hydroxylation of kynurenine to form 3-hydroxykynurenine. This enzyme has been reported to be involved in the generation of oxidative radicals as well as in cytokine-mediated inflammation [57]. Leucine amino peptidase 3 (LAP3) is involved

**Table 2. The differentially expressed genes (DEGs) of mitochondrial proteins in at least three synovial microarray datasets.**

| S.No. | Gene | Number of RA synovial datasets in which the gene was differentially expressed | | | | Max fold-change | |
|---|---|---|---|---|---|---|---|
| | | Up-regulated | Down-regulated | Total | Type of regulation | Linear | log base 2 |
| 1 | AK4 | 1 | 2 | 3 | Mixed | 1.87 | 0.90 |
| 2 | AKR1B10 | 0 | 3 | 3 | Down | 0.26 | -1.94 |
| 3 | BCL2 | 3 | 1 | 4 | Mixed | 0.38 | -1.39 |
| 4 | C10orf10 | 1 | 2 | 3 | Mixed | 0.17 | -2.55 |
| 5 | DNAJC15 | 3 | 0 | 3 | Up | 1.72 | 0.78 |
| 6 | IDH2 | 3 | 0 | 3 | Up | 3.66 | 1.87 |
| 7 | MAOA | 0 | 3 | 3 | Down | 0.09 | -3.47 |
| 8 | MCCC1 | 0 | 3 | 3 | Down | 0.58 | -0.78 |
| 9 | PDK4 | 0 | 3 | 3 | Down | 0.12 | -3.05 |
| 10 | YME1L1 | 3 | 0 | 3 | Up | 3.22 | 1.68 |
| 11 | PRDX4 | 3 | 0 | 3 | Up | 4.62 | 2.20 |
| 12 | UCP2 | 4 | 0 | 4 | Up | 7.94 | 2.98 |
| 13 | C10orf2 | 0 | 3 | 3 | Down | 0.61 | -0.71 |
| 14 | ACOT7 | 5 | 0 | 5 | Up | 2.75 | 1.45 |
| 15 | EFHD1 | 0 | 3 | 3 | Down | 0.26 | -1.94 |
| 16 | IFI27 | 4 | 0 | 4 | Up | 3.54 | 1.82 |
| 17 | KMO | 5 | 0 | 5 | Up | 4.55 | 2.18 |
| 18 | PLGRKT | 3 | 0 | 3 | Up | 2.59 | 1.37 |
| 19 | SLC16A7 | 2 | 4 | 6 | Mixed | 0.28 | -1.83 |
| 20 | CASP8 | 3 | 0 | 3 | Up | 2.4 | 1.26 |
| 21 | LAP3 | 5 | 0 | 5 | Up | 2.73 | 1.44 |
| 22 | PDK1 | 4 | 0 | 4 | Up | 4.81 | 2.26 |
| 23 | C15orf48 | 3 | 0 | 3 | Up | 30.45 | 4.92 |

The number of datasets in which the gene was up/down-regulated is also given in the table along with the maximum observed fold-change of the genes among the datasets.

in the degradation of glutathione, a scavenger of free radicals [58], indicating the likely impairment in the detoxification of ROS. Pyruvate dehydrogenase kinase 1(PDK1) inhibits pyruvate dehydrogenase activity and is known to play an integral role against hypoxia- and oxidative stress-mediated apoptosis [59]. Interferon alpha inducible protein 27 (IFI27) is known to be involved in cytokine signalling and apoptosis. Interestingly, this protein activates an apoptotic caspase, CASP8, which was also up-regulated in the microarray data [60]. Uncoupling protein 2 (UCP2) is implicated in the transfer of anions from the inner to the outer membrane and protons from the outer to the inner membrane, and it is known to control ROS [61]. Peroxiredoxin-4 (PRDX4) is an antioxidant enzyme which detoxifies $H_2O_2$ and regulates NF-κB activation [62]. Nonetheless, because of its high reactivity, this enzyme is susceptible to overoxidation and inactivation by $H_2O_2$ [63]. YME1 like 1 ATPase (YME1L1), which is an ATP-dependent metalloprotease, is known to function in the maintenance of mitochondrial morphology and accumulation of respiratory chain subunits [64–65]. Isocitrate dehydrogenase 2 (IDH2) is implicated in the production of NADPH and in the protection of cells from ROS. DnaJ heat shock protein family (Hsp40) member C15 (DNAJC15) negatively regulates respiratory chain and generation of ATP.

Similarly, the six genes that were down-regulated in at least three datasets participate in the following functions. MCCC1 encodes α subunit of 3-methylcrotonoyl-CoA carboxylase (3-MCC), which is an enzyme that is involved in the breakdown of leucine. Monoamine oxidase A (MAOA) catalyses the oxidative deamination of amines, such as serotonin, norepinephrine and dopamine, and its deficiency is known to induce aggression [66–67]. The transcription factors, specificity protein 1 (SP1), GATA binding protein 2 (GATA2) and TATA box binding protein (TBP) regulate the expression of this gene [68]. EF-hand domain-containing protein 1 (EFHD1) is a calcium ion sensor. Some of the other down-regulated genes are AKR1B10, PDK4 and C10orf2.

The mixed-regulated gene, solute carrier family 16 member 7 (SLC16A7) is involved in the transport of metabolites such as monocarboxylates and pyruvate. Similarly, adenylate kinase 4 (AK4) is implicated in the metabolism of nucleotides.

The largest genome-wide association study meta-analysis of RA cases and controls has identified 98 disease risk genes [69]. Six of them, C1QBP, SUOX, ACSL6, UNG, CYP27B1 and CASP8 are MitoCarta genes. Among these, only CASP8 was a DEG, and C1QBP and CYP27B1 are part of the created mitochondrial network. The role of these genes in RA and their involvement in mitochondrial dysfunction remain to be ascertained.

## Effects of medical therapies on gene expression

Of the six open-source microarray datasets we analysed, RA patients in one (GSE7307) were not treated with therapies, while the patients belonging to three others (GSE12021 (HGU133A), GSE12021 (HGU133B) and GSE55457) underwent different combinations of medical therapies. The information on medications is not available for two datasets (GSE55235 and GSE77298). All the details of medical therapies available for the datasets are listed in Table 3.

It is seen within a dataset that some patients have received the same combination of medical therapies whereas others received different combinations. To test if the gene expressions are influenced by these medical therapies, the samples in each microarray dataset were hierarchically clustered based on the mRNA levels of the DEGs that were differentially expressed in at least three microarray datasets (Table 2). The cluster results are shown as heatmaps with dendrograms (S2–S7 Figs). In GSE7307, GSE55235 and GSE55457, RA and control samples were clustered into separate groups (S2–S4 Figs). In GSE12021 (HGU133A) and GSE12021

**Table 3. Medical therapies initiated on RA patients that participated in the microarray studies.**

| Dataset | Patients | Medical Therapies |
|---|---|---|
| GSE7307 | | All the patients were not treated |
| GSE12021A | RA1 | NSARD + Azulfidine + Prednisolone |
| | RA2 | NSARD + MTX + Prednisolone |
| | RA3 | NSARD + MTX+ Prednisolone |
| | RA4 | NSARD + Azulfidine + Prednisolone + MTX |
| | RA5 | NSARD + MTX + Prednisolone |
| | RA6 | NSARD + Azulfidine + Prednisolone |
| | RA7 | MTX + Prednisolone |
| | RA8 | NSARD |
| | RA9 | NSARD + Prednisolone |
| | RA10 | NSARD + Prednisolone |
| | RA11 | COX-2 inhibitor + Prednisolone + Quensyl |
| | RA12 | NSAID + Tilidin + Prednisolone |
| GSE12021B | RA1 | NSARD + Azulfidine + Prednisolone |
| | RA2 | NSARD + MTX + Prednisolone |
| | RA3 | NSARD + MTX+ Prednisolone |
| | RA4 | NSARD + Azulfidine + Prednisolone + MTX |
| | RA5 | NSARD + MTX + Prednisolone |
| | RA6 | NSARD + Azulfidine + Prednisolone |
| | RA7 | MTX + Prednisolone |
| | RA8 | NSARD |
| | RA9 | NSARD + Prednisolone |
| | RA10 | NSARD + Prednisolone |
| | RA11 | COX-2 inhibitor + Prednisolone + Quensyl |
| | RA12 | NSAID + Tilidin + Prednisolone |
| GSE55457 | RA1 | NSARD + Azulfidine + Prednisolone |
| | RA2 | NSARD + MTX + Prednisolone |
| | RA3 | NSARD + MTX+ Prednisolone |
| | RA4 | NSARD + Azulfidine + Prednisolone + MTX |
| | RA5 | NSARD + MTX + Prednisolone |
| | RA6 | NSARD + Azulfidine + Prednisolone |
| | RA7 | MTX + Prednisolone |
| | RA8 | NSARD |
| | RA9 | NSARD + Prednisolone |
| | RA10 | no therapy used |
| | RA11 | NSARD + Prednisolone |
| | RA12 | COX-2 inhibitor + Prednisolone + Quensyl |
| | RA13 | NSAID + Tilidin + Prednisolone |
| GSE55235 | | Therapies not mentioned for these two datasets |
| GSE77298 | | |

NSARD, nonsteroidal anti-rheumatic drug; MTX, methotrexate; COX-2, cyclooxygenase-2; NSAID, nonsteroidal anti-inflammatory drug

(HGU133B), some RA samples were clustered into a separate group while others were clustered with control samples (S5 and S6 Figs), showing that there is a drug effect. In GSE77298, some RA samples were clustered with healthy controls but drug therapies are not available for this dataset (S7 Fig).

In order to find the effect of medical therapies on the differential expression of genes, we removed the RA samples that were clustered with healthy controls from GSE12021 (HGU133A) and GSE12021 (HGU133B) datasets and repeated the differential expression analysis for the 23 genes listed in Table 2. Surprisingly, with the same selection criteria of differential expression, all the 23 genes were retained. The heatmaps of the expression levels of the genes in these two datasets after eliminating the RA samples that clustered with healthy controls are shown in S8 and S9 Figs. We notice the complete separation of controls from RA samples in the clusters.

From the above analysis, we find that the 23 genes were differentially expressed in at least three datasets in both of the following cases.

Case 1: all RA and control samples in all the six studies

Case 2: all RA samples except those that clustered with healthy controls in GSE12021 (HGU133A) and GSE12021 (HGU133B) and all controls in all the six studiesSince the 23 genes were differentially expressed in at least three datasets in both the cases, we conclude that these genes were not affected by the therapy initiation.

In addition to the above analysis, we analysed two other microarray datasets (GSE77344 and GSE11237) where patients with diseases unrelated to RA were treated with prednisone or celecoxib. Prednisone is the prodrug form of prednisolone, while celecoxib is a COX-2 inhibitor. Prednisolone and COX-2 inhibitors are part of the therapies received by RA patients shown in Table 3. In the dataset GSE77344 [70], whole blood was collected from patients with chronic obstructive pulmonary disease who were either treated (n = 31) or not treated (n = 103) with prednisone. GSE11237 [71] contained colorectal primary adenocarcinomas surgically removed from 23 patients, 11 of whom received 400 mg celecoxib two times per day for seven days prior to surgery and 12 who did not receive the treatment. In addition to this, we also analysed GSE45867 [72] which had paired synovial tissue biopsies from 8 early RA patients naive to methotrexate or DMARDs. The samples were collected before and 12 weeks after the initiation of methotrexate therapy. Hierarchical clustering of the samples based on the expression levels of the 23 genes normalized across samples did not show any separation of treated and nontreated samples, or pre and post treatment samples (S10–S12 Figs). Differential expression analysis with the criteria used for the RA datasets (fold change $> |1.5|$ and p value $< = 0.05$) revealed one gene out of the 23 was upregulated in GSE77344 (MAOA, fold change = 3.9, pvalue = 0.001), while no differential regulation was found for any of the 23 genes in GSE11237 and GSE45867. Thus we believe that the effects of these specific treatments on the candidate genes are negligible, and the differential regulation observed in the RA datasets is more likely due to the disease itself.

## Identification of hubs of the PPI network

To further elucidate the properties of the mitochondrial PPI network, we performed network analysis. For each node, we chose to measure the network parameter 'degree' which is the number of edges a node can have. The probability distribution of the degree of nodes in the created mitochondrial PPI network along with power-law fit to the data is shown in Fig 2. The degree distribution of the network follows a power law $P(k) \sim k^{-\alpha}$ (with the degree coefficient, $\alpha = 1.82$), which is a property of scale-free networks [73]. From this network, we identify a small number of important nodes, called hubs, which are directly connected to a large number of interacting partners. Analogous to social networks, the hub proteins with a higher number of neighbours are crucial to PPI networks as their removal causes dysfunction of the system.

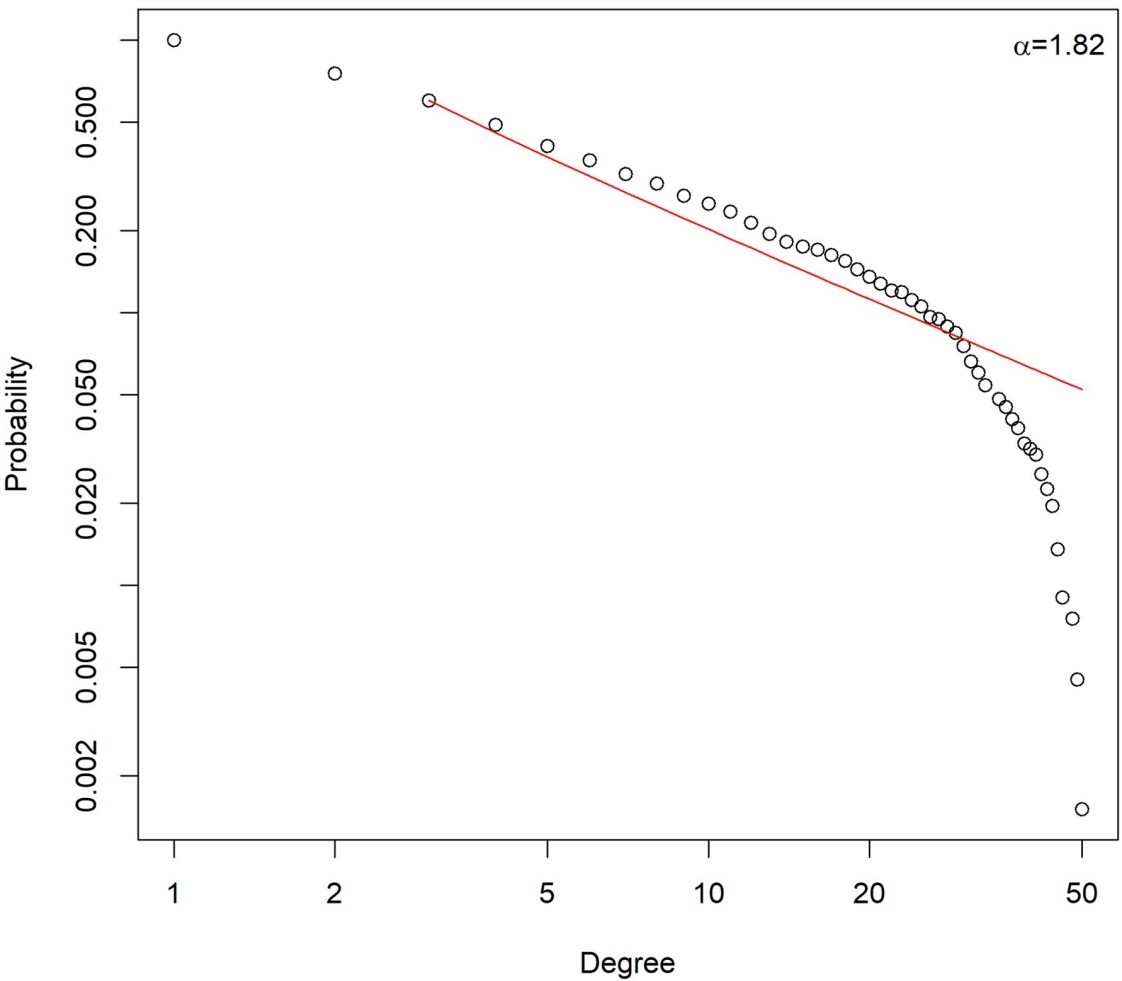

**Fig 2. Degree distribution of the mitochondrial PPI network (nodes: 665, edges: 2708), following a power law.** The circles represent the fraction of nodes with a given degree and the solid line indicates the power-law fit to the data.

The immediate neighbours of all the 665 proteins in the mitochondrial PPI network were determined. The top 50 proteins in the decreasing order of the number of their immediate neighbours are listed in Table 4. The entire list of all the network proteins and the number of their immediate neighbours—including the extent of DEGs among them—could be found in S3 Table. The distributions of the proteins in terms of the total number of neighbours and the proportion of DEGs among them are shown in S13 Fig.

Each of the network proteins has at least one neighbour. Among them, 167 have at least 10 neighbours. Most of the network proteins are connected to one or a few DEGs. The scatter plot between the number of neighbours and the number of DEGs among the neighbours for individual proteins is shown in Fig 3. It would be interesting to look at the hubs with a high number of neighbours containing higher number of DEGs among them. For example, the upper right-side rectangle of the figure has the hubs connected to at least 27 neighbours having a minimum of seven DEGs among them. The hubs which have fallen into this rectangle are given in Table 5. They could be considered crucial for mitochondrial functions in the RA diseased synovium because of a high number of DEGs among the neighbours.

**Table 4. Number of neighbours for the top 50 mitochondrial PPI network hub proteins.**

| S.No. | Protein | Neighbours | DEGs | Up DEG | Down DEG | mixed DEGs |
|---|---|---|---|---|---|---|
| 1 | UQCR10 | 50 | 9 | 2 | 6 | 1 |
| 2 | MRPL4 | 49 | 4 | 0 | 4 | 0 |
| 3 | NDUFV2 | 49 | 8 | 0 | 7 | 1 |
| 4 | UQCRC2 | 48 | 6 | 0 | 5 | 1 |
| 5 | UQCRQ | 48 | 9 | 3 | 5 | 1 |
| 6 | NDUFS3 | 46 | 5 | 1 | 4 | 0 |
| 7 | MRPL47 | 45 | 4 | 0 | 4 | 0 |
| 8 | NDUFA13 | 45 | 8 | 2 | 5 | 1 |
| 9 | NDUFB8 | 45 | 7 | 2 | 4 | 1 |
| 10 | ATP5O | 44 | 8 | 2 | 5 | 1 |
| 11 | MRPL24 | 44 | 3 | 0 | 3 | 0 |
| 12 | NDUFS6 | 44 | 11 | 4 | 6 | 1 |
| 13 | UQCRFS1 | 44 | 7 | 2 | 4 | 1 |
| 14 | CYC1 | 43 | 6 | 2 | 4 | 0 |
| 15 | NDUFAB1 | 43 | 8 | 1 | 6 | 1 |
| 16 | MRPL13 | 42 | 5 | 0 | 5 | 0 |
| 17 | MRPL16 | 42 | 2 | 0 | 2 | 0 |
| 18 | ATP5C1 | 41 | 7 | 1 | 5 | 1 |
| 19 | MRPS16 | 41 | 2 | 2 | 0 | 0 |
| 20 | NDUFB10 | 41 | 7 | 1 | 5 | 1 |
| 21 | NDUFA9 | 40 | 7 | 2 | 4 | 1 |
| 22 | MRPL15 | 39 | 6 | 0 | 6 | 0 |
| 23 | COX5B | 38 | 7 | 2 | 4 | 1 |
| 24 | NDUFA8 | 38 | 4 | 0 | 3 | 1 |
| 25 | UQCRC1 | 38 | 5 | 2 | 3 | 0 |
| 26 | COX6A1 | 37 | 2 | 0 | 2 | 0 |
| 27 | NDUFA2 | 37 | 7 | 2 | 5 | 0 |
| 28 | MRPL3 | 36 | 5 | 0 | 5 | 0 |
| 29 | NDUFB9 | 36 | 4 | 2 | 2 | 0 |
| 30 | SDHB | 36 | 5 | 0 | 4 | 1 |
| 31 | NDUFA6 | 35 | 6 | 3 | 2 | 1 |
| 32 | UQCRB | 35 | 8 | 2 | 5 | 1 |
| 33 | MRPS9 | 33 | 3 | 0 | 3 | 0 |
| 34 | NDUFB2 | 33 | 6 | 1 | 5 | 0 |
| 35 | NDUFB6 | 33 | 5 | 0 | 5 | 0 |
| 36 | NDUFS2 | 33 | 3 | 1 | 2 | 0 |
| 37 | ATP5B | 32 | 5 | 2 | 3 | 0 |
| 38 | ATP5L | 32 | 6 | 2 | 3 | 1 |
| 39 | MRPL39 | 32 | 2 | 0 | 2 | 0 |
| 40 | NDUFB4 | 32 | 4 | 0 | 4 | 0 |
| 41 | MRPS30 | 31 | 2 | 0 | 2 | 0 |
| 42 | NDUFA1 | 31 | 6 | 1 | 5 | 0 |
| 43 | NDUFB11 | 31 | 4 | 0 | 3 | 1 |
| 44 | TUFM | 31 | 0 | 0 | 0 | 0 |
| 45 | MRPL12 | 30 | 2 | 1 | 1 | 0 |
| 46 | MRPL17 | 30 | 3 | 1 | 2 | 0 |
| 47 | MRPL19 | 30 | 5 | 0 | 5 | 0 |

*(Continued)*

**Table 4.** (Continued)

| S.No. | Protein | Neighbours | DEGs | Up DEG | Down DEG | mixed DEGs |
|---|---|---|---|---|---|---|
| 48 | MRPL27 | 30 | 2 | 1 | 1 | 0 |
| 49 | MRPL40 | 30 | 2 | 0 | 2 | 0 |
| 50 | SDHA | 30 | 2 | 0 | 2 | 0 |

The table shows the number of first neighbours, the number of DEGs, and number of up/down-regulated DEGs among the first neighbours.

We chose four representative hubs from the top right-side rectangle to draw their subnetworks with immediate connecting proteins (points marked red in the scatter plot, Fig 3). One of these hubs, UQCR10, which is a subunit of the respiratory chain complex III, is connected to 50 proteins, including two up- and six down-regulated DEGs (Table 5). The DEGs include the up-regulated complex 1 subunit NDUFB7 and complex III subunit UQCR11; the down-

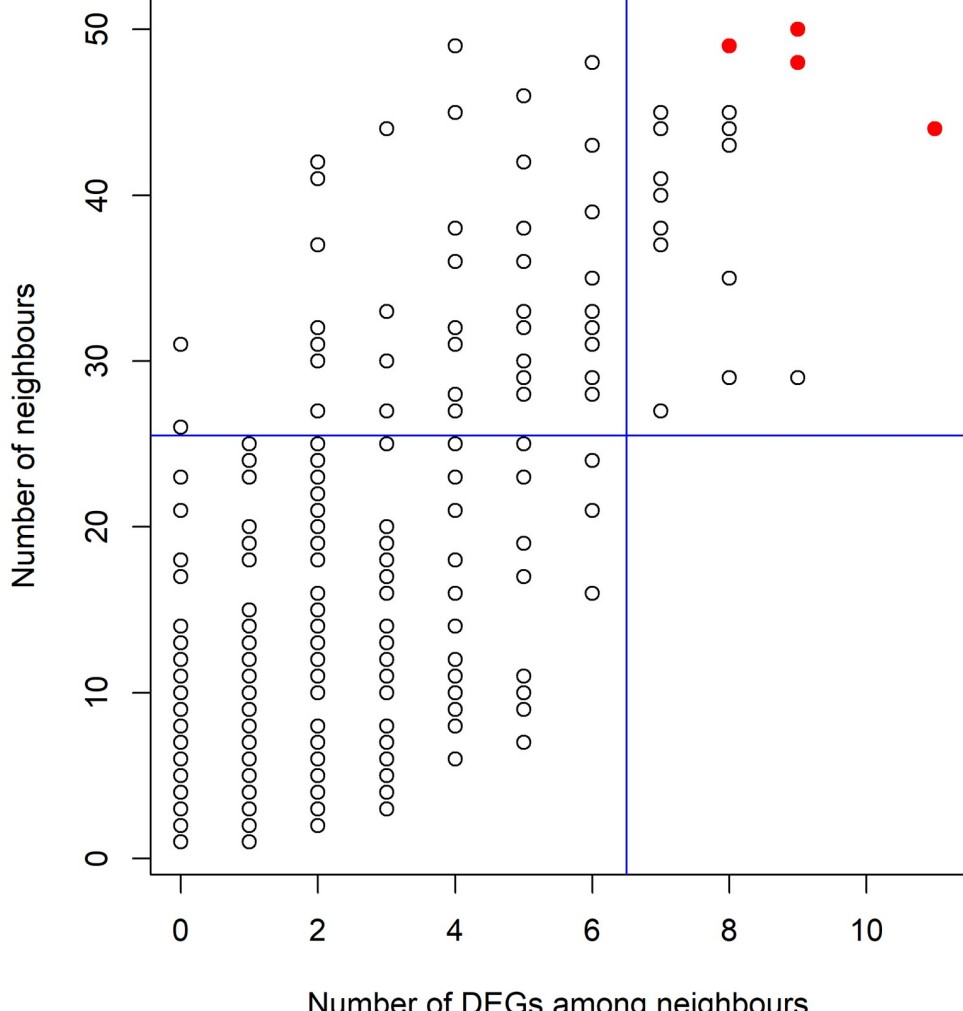

**Fig 3. The scatterplot showing the relation between the number of neighbours and DEGs among them.** The right top rectangle shows hub proteins with a high number of neighbours as well as DEGs among neighbours. The four circles represented in red colour correspond to NDUFS6, UQCR10, UQCRQ and NDUFV2.

**Table 5. The important hubs with a high number of neighbours and DEGs among them.**

| S.No. | Protein | Neighbours | DEGs | Up DEG | Down DEG | Mixed DEGs |
|---|---|---|---|---|---|---|
| 1 | NDUFS6 | 44 | 11 | 4 | 6 | 1 |
| 2 | UQCR10 | 50 | 9 | 2 | 6 | 1 |
| 3 | UQCRQ | 48 | 9 | 3 | 5 | 1 |
| 4 | ACLY | 29 | 9 | 5 | 4 | 0 |
| 5 | NDUFV2 | 49 | 8 | 0 | 7 | 1 |
| 6 | NDUFA13 | 45 | 8 | 2 | 5 | 1 |
| 7 | ATP5O | 44 | 8 | 2 | 5 | 1 |
| 8 | NDUFAB1 | 43 | 8 | 1 | 6 | 1 |
| 9 | UQCRB | 35 | 8 | 2 | 5 | 1 |
| 10 | NDUFA12 | 29 | 8 | 2 | 5 | 1 |
| 11 | NDUFB8 | 45 | 7 | 2 | 4 | 1 |
| 12 | UQCRFS1 | 44 | 7 | 2 | 4 | 1 |
| 13 | ATP5C1 | 41 | 7 | 1 | 5 | 1 |
| 14 | NDUFB10 | 41 | 7 | 1 | 5 | 1 |
| 15 | NDUFA9 | 40 | 7 | 2 | 4 | 1 |
| 16 | COX5B | 38 | 7 | 2 | 4 | 1 |
| 17 | NDUFA2 | 37 | 7 | 2 | 5 | 0 |
| 18 | ATP5H | 27 | 7 | 3 | 2 | 2 |

regulated complex IV subunit COX7A1, complex I subunits, NDUFA4, NDUFB4, NDUFB6 and NDUFB9, and complex III subunit UQCRFS1 (S14 Fig). Another hub, NDUFV2 is linked to 49 proteins, including eight DEGs, seven of which, namely NDUFA4, NDUFB4, NDUFB6, NDUFB9, NDUFS4, PITRM1 and UQCRFS1, were down-regulated and one gene was mixed-regulated (S15 Fig).

The third hub, NDUFS6 links to 11 DEGs: four of these were up-regulated and are part of complex III (UQCR11), complex I (NDUFB7), complex IV (COX15) and complex V (ATP5E); six were down-regulated, of which ECHDC2 is implicated in the lyase activity, and the rest (NDUFA4, NDUFB4, NDUFB6, NDUFB9 and NDUFS4) are complex 1 subunits (S16 Fig).

The hub protein UQCRQ links to nine neighbours, of which three were up-regulated (UQCR11, NDUFB7 and ATP5E); and five were down-regulated (NDUFA4, NDUFB4, NDUFB6, NDUFS4 and UQCRFS1) (S17 Fig).

## Gene ontology (GO) and pathway enrichment of the mitochondrial DEGs

The GO and pathway analyses were performed for the mitochondrial DEGs that were selected in at least two datasets (83 genes), using the STRING database. The results illustrated their roles at different levels, including MF, BP and CC. A total of 14 MF, 63 BP and 16 CC GO terms, and 3 KEGG pathways were significantly enriched (FDR < 0.01). Several of these terms and pathways were shared by both the up- and down-regulated DEGs. Some of the significantly enriched BP and MF GO terms are shown in Fig 4. The detailed lists of all the GO terms and KEGG pathways could be found in S4 Table and Tables 6–8.

Among the BP terms, the processes pertaining to metabolism, mitochondrial membrane permeability, regulation of membrane potential, oxidative stress, mitochondrial transport and apoptotic processes were enriched with DEGs (Fig 4 and S4 Table). Similarly, among the MF terms, the functions related to the binding of coenzymes and cofactors were enormously

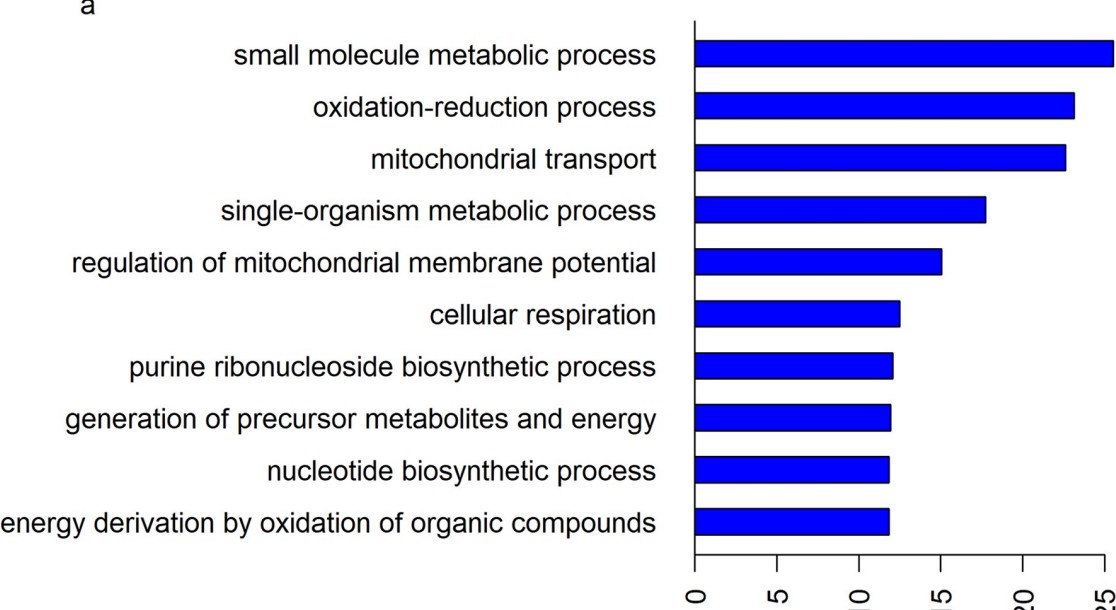

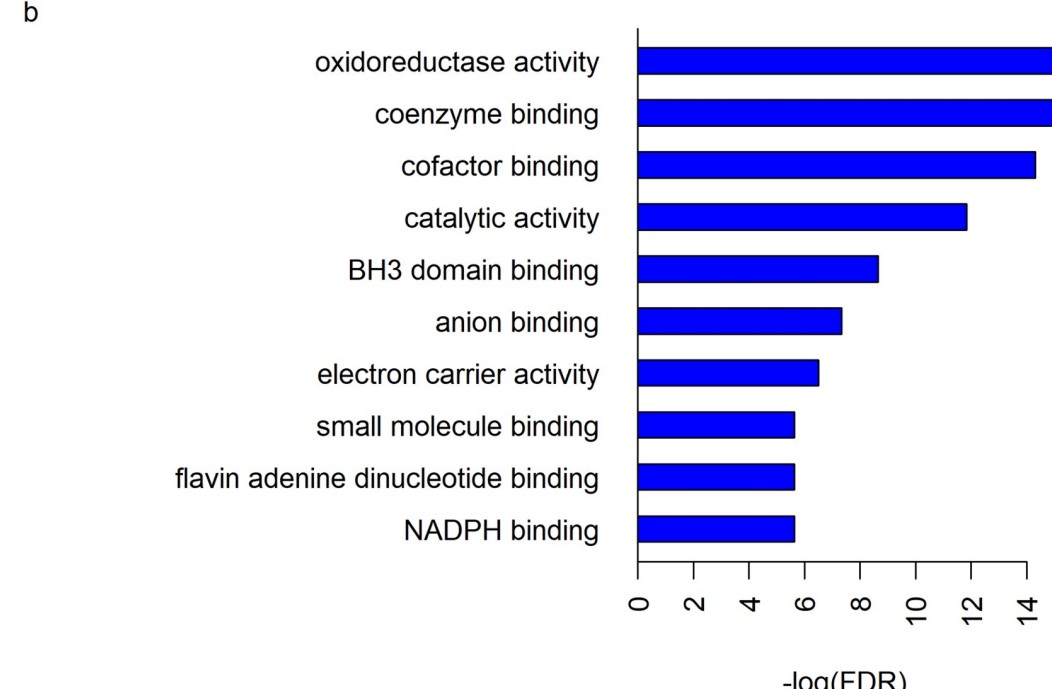

**Fig 4. Some of the significantly enriched (a) BP and (b) MF GO terms.** Each GO term was plotted against the negative logarithm of its false discovery rate (FDR) obtained from GO analysis using the STRING database, which uses hypergeometric test for determining significantly enriched GO terms [55–56].

**Table 6. The significantly enriched molecular functions (MF) of the RA synovial mitochondrial DEGs.**

| S.No. | Pathway ID | Pathway description | Observed gene count | False discovery rate |
|---|---|---|---|---|
| 1 | GO.0016491 | oxidoreductase activity | 18 | 2.85E-07 |
| 2 | GO.0050662 | coenzyme binding | 11 | 3.06E-07 |
| 3 | GO.0048037 | cofactor binding | 12 | 6.02E-07 |
| 4 | GO.0003824 | catalytic activity | 44 | 7.19E-06 |
| 5 | GO.0051434 | BH3 domain binding | 3 | 0.000176 |
| 6 | GO.0043168 | anion binding | 27 | 0.000651 |
| 7 | GO.0009055 | electron carrier activity | 6 | 0.00149 |
| 8 | GO.0036094 | small molecule binding | 25 | 0.00356 |
| 9 | GO.0050660 | flavin adenine dinucleotide binding | 5 | 0.00356 |
| 10 | GO.0070402 | NADPH binding | 3 | 0.00356 |
| 11 | GO.0000166 | nucleotide binding | 23 | 0.00471 |
| 12 | GO.0046899 | nucleoside triphosphate adenylate kinase activity | 2 | 0.00471 |
| 13 | GO.0022857 | transmembrane transporter activity | 13 | 0.0067 |
| 14 | GO.0050661 | NADP binding | 4 | 0.00833333 |

**Table 7. The significantly enriched cellular components (CC) of the RA synovial mitochondrial DEGs.**

| S.No. | Pathway ID | Pathway description | Observed gene count | False discovery rate |
|---|---|---|---|---|
| 1 | GO.0005739 | Mitochondrion | 61 | 1.16E-47 |
| 2 | GO.0044429 | mitochondrial part | 43 | 5.65E-35 |
| 3 | GO.0005740 | mitochondrial envelope | 38 | 2.20E-32 |
| 4 | GO.0031966 | mitochondrial membrane | 35 | 5.41E-29 |
| 5 | GO.0031967 | organelle envelope | 39 | 6.11E-27 |
| 6 | GO.0019866 | organelle inner membrane | 27 | 3.67E-21 |
| 7 | GO.0005743 | mitochondrial inner membrane | 26 | 7.18E-21 |
| 8 | GO.0005741 | mitochondrial outer membrane | 14 | 1.49E-13 |
| 9 | GO.0005759 | mitochondrial matrix | 17 | 6.25E-12 |
| 10 | GO.0031090 | organelle membrane | 39 | 2.37E-11 |
| 11 | GO.0044444 | cytoplasmic part | 58 | 5.34E-10 |
| 12 | GO.0005737 | Cytoplasm | 59 | 9.81E-05 |
| 13 | GO.0044446 | intracellular organelle part | 49 | 0.000261 |
| 14 | GO.0044455 | mitochondrial membrane part | 7 | 0.000336 |
| 15 | GO.0043231 | intracellular membrane-bounded organelle | 58 | 0.000913 |
| 16 | GO.0097136 | Bcl-2 family protein complex | 2 | 0.00794 |

enriched with DEGs (Fig 4 and Table 6). Additionally, DEGs were enriched in many mitochondrial CC terms (Table 7).

Among the enriched KEGG pathways, 'metabolic pathways' (KEGG pathway ID: 1100) is highly enriched with 20 DEGs. Of the others, glycine, serine, threonine and glutathione metabolisms were affected (Table 8).

**Table 8. The significantly enriched KEGG pathways of the RA synovial mitochondrial DEGs.**

| S.No. | Pathway ID | Pathway description | Observed gene count | False discovery rate |
|---|---|---|---|---|
| 1 | 1100 | Metabolic pathways | 20 | 7.81E-06 |
| 2 | 260 | Glycine, serine and threonine metabolism | 5 | 7.21E-05 |
| 3 | 480 | Glutathione metabolism | 4 | 0.00444 |

## Disruption of OxPhos in the RA synovium

From the enriched MF items, it is understood that the oxidoreductase activity is getting affected in RA. This activity is associated with the OxPhos complexes of mitochondria. There are five complexes: complex I, II, III, IV and V. Electrons from NADH and FADH2 pass through the first four complexes and eventually reduce $O_2$ to water at complex IV. Overall, the complexes have 97 subunits, 84 of which are encoded by nDNA. The created mitochondrial PPI network contains 81 of the 84 subunits. Totally, 11 of them were DEGs in one or two datasets (Table 9). The complex1 DEGs, NDUFB4, NDUFB6, NDUFB9 and NDUFS4 were down-regulated. At least one gene each in complex III, complex IV and complex V was down or up-regulated. UQCRFS1 of complex III, COX6A1 and COX7A1 of complex IV, and ATP5G3 of complex V were down-regulated. The mitochondrial protein-coding genes of OxPhos were either missed or non-DEGs in the microarray studies. From these observations, it can be deduced that OxPhos is getting disrupted, perhaps impaired due to the decreased expression of some of the OxPhos genes in the RA synovium. This might be concerned with the escape of electrons from OxPhos, leading to the formation of ROS. Nevertheless, since 11 of the subunits are DEGs in either one or two datasets, it may be difficult to come to a concrete conclusion.

## ROS detoxification and apoptosis in the RA synovium

The ROS generating NADPH oxidases (NOXs), such as NOX1, NOX2 and NOX3 were not DEGs, and NOX4 was down in two and up in one microarray datasets. This might be indicating that it is the mitochondria, and not these enzymes, that could be the primary source of ROS in RA. The detoxifiers of ROS, such as catalase (CAT), glutathione peroxidase 4 (GPX4) and superoxide dismutase 1 (SOD1) were down-regulated in one dataset, specifying impaired detoxification. Further, LAP3, which degrades glutathione, was overexpressed in five datasets. This gives a clue for the accumulation of mitochondrial ROS (mtROS), which might induce apoptosis in RA synoviocytes. In support of this, the pro-apoptotic BAX was up-regulated in two datasets. The executor of apoptosis, CASP8 was up in three. However, CASP8, in the presence of FLIP, is known to induce inflammation rather than apoptosis [45]. Further, $H_2O_2$ might trigger apoptosis by causing elevation of intracellular $Ca^{2+}$ levels via a pathway that includes spleen tyrosine kinase (SYK), Bruton's tyrosine kinase (BTK), the B cell linker protein (BLNK) and phospholipase Cγ2 (PLCγ2) [74–75]. Interestingly, these four genes were up-

**Table 9. The differential expression of the subunits of mitochondrial respiratory chain complexes and their maximum fold-changes.**

| S. No. | Gene | Number of synovial datasets with up-regulation | Number of synovial datasets with down-regulation | Mitochondrial respiratory chain complex | Max fold-change |
|---|---|---|---|---|---|
| 1 | NDUFB4 | 0 | 1 | I | 1.62 ↓ |
| 2 | NDUFB6 | 0 | 1 | I | 1.7 ↓ |
| 3 | NDUFB7 | 1 | 0 | I | 1.76 ↑ |
| 4 | NDUFB9 | 0 | 1 | I | 1.76 ↓ |
| 5 | NDUFS4 | 0 | 1 | I | 2.15 ↓ |
| 6 | UQCRFS1 | 0 | 1 | III | 1.69 ↓ |
| 7 | UQCR11 | 1 | 0 | III | 1.51 ↑ |
| 8 | COX6A1 | 1 | 1 | IV | 1.63 ↑ |
| 9 | COX7A1 | 0 | 2 | IV | 2.94 ↓ |
| 10 | ATP5E | 2 | 0 | V | 1.58 ↑ |
| 11 | ATP5G3 | 0 | 1 | V | 1.73 ↓ |

The up and down arrows indicate up and down-regulation, respectively.

regulated in at least three datasets. SYK, BTK, BLNK and PLCγ2 were up-regulated in six, three, five and six datasets, respectively. But, Bcl-2-like 1 (BCL2L1) protein, which controls the production of ROS by regulating membrane potential, was mixed-regulated, making it difficult to conclude on its role. Further, the anti-apoptotic protein Bcl-2 (BCL2) was mixed-regulated. Surprisingly, the pro-apoptotic Bcl-2-like 13 (BCL2L13) was down in one dataset. Collectively, the results may suggest (i) generation of mtROS, (ii) impaired ROS detoxification and (iii) induction of mitochondrion-initiated intrinsic apoptosis in the RA synovium.

## A model relating mtROS and inflammation in the RA synovium

Since NOXs are less expressed, mitochondria might be the primary generators of ROS in the RA synovium. A component of ROS, $O_2 \cdot^-$ may interact with nitric oxide (NO), which is abundant in RA [76–81], resulting in the formation of $ONOO^-$. The molecule $ONOO^-$ is involved in the activation of the IκB kinase (IKK) through a mechanism that depletes the–SH groups of glutathione [43]. Our analysis also indicated an increased degradation of glutathione, possibly by the enzyme LAP3. Active IKK degrades its substrate IκB, resulting in the translocation of NF-κB to the nucleus, where it induces the expression of inflammatory mediators, such as TNF, IL-1β and iNOS. The enzyme iNOS catalyses the production of NO from arginine and hence might further potentiate the production of $ONOO^-$ (Fig 5). Furthermore, damaged mitochondria can release molecules, called the damage-associated molecular patterns (DAMPs), which contribute to inflammation [82]. Taken together, mitochondrial production of ROS and their involvement in NF-κB activation, increased glutathione degradation, and DAMPs released from damaged mitochondria suggest that this cell organelle is associated with the induction and maintenance of inflammation in the RA synovium.

## Discussion

Many researchers had created PPI networks for RA at the level of a cell. For instance, in order to identify key molecules, earlier we had created a PPI network for cytokine signalling in RA [83]. Similarly, few studies had identified highly connected regions, ego networks and key genes in PPI networks in RA and other diseases [84–86]. Another study used PPI for determining the efficacy of leflunomide and ligustrazine drugs in the treatment of RA [87]. In contrast, in this study, we created, for the first time, a PPI network that is specific to RA synovial mitochondria and identified hub proteins. This network was created using reliable interactions from seven databases and gene expression data from six open-source microarray datasets. The network has 665 genes, of which 131 are DEGs.

The GO analysis of DEGs has given enriched processes, functions, cell components and KEGG pathways. 'Oxidoreductase activity' is the highest enriched molecular function. In general, several metabolic mechanisms, including OxPhos and pathways related to nucleotide, amino acid and glutathione seem to be affected. The analysis also identified the enriched oxidative stress, electron carrier activity, membrane permeability and apoptosis-related GO terms with DEGs.

The analysis of six microarray datasets gave 208 mitochondrial DEGs. Among them, the up-regulation of IFI27, which is involved in cytokine-mediated apoptosis, is in agreement with other RA studies [88]. The enzyme PDK1 was up-regulated. It is involved in hypoxia- and oxidative stress-mediated apoptosis, and is known to induce Akt pathway in human mast cells—which are abundantly seen in the RA synovium [89]. This enzyme also induces cell invasion and secretion of IL-1β and IL-6 in a ribosomal S6 kinase (RSK2)-dependent TNF pathway in FLS [90]. AKR1B10 was down-regulated in three datasets. There is evidence that hypoxia induces the down-regulation of this gene in RA and healthy FLS [91]. EFHD1, which is a

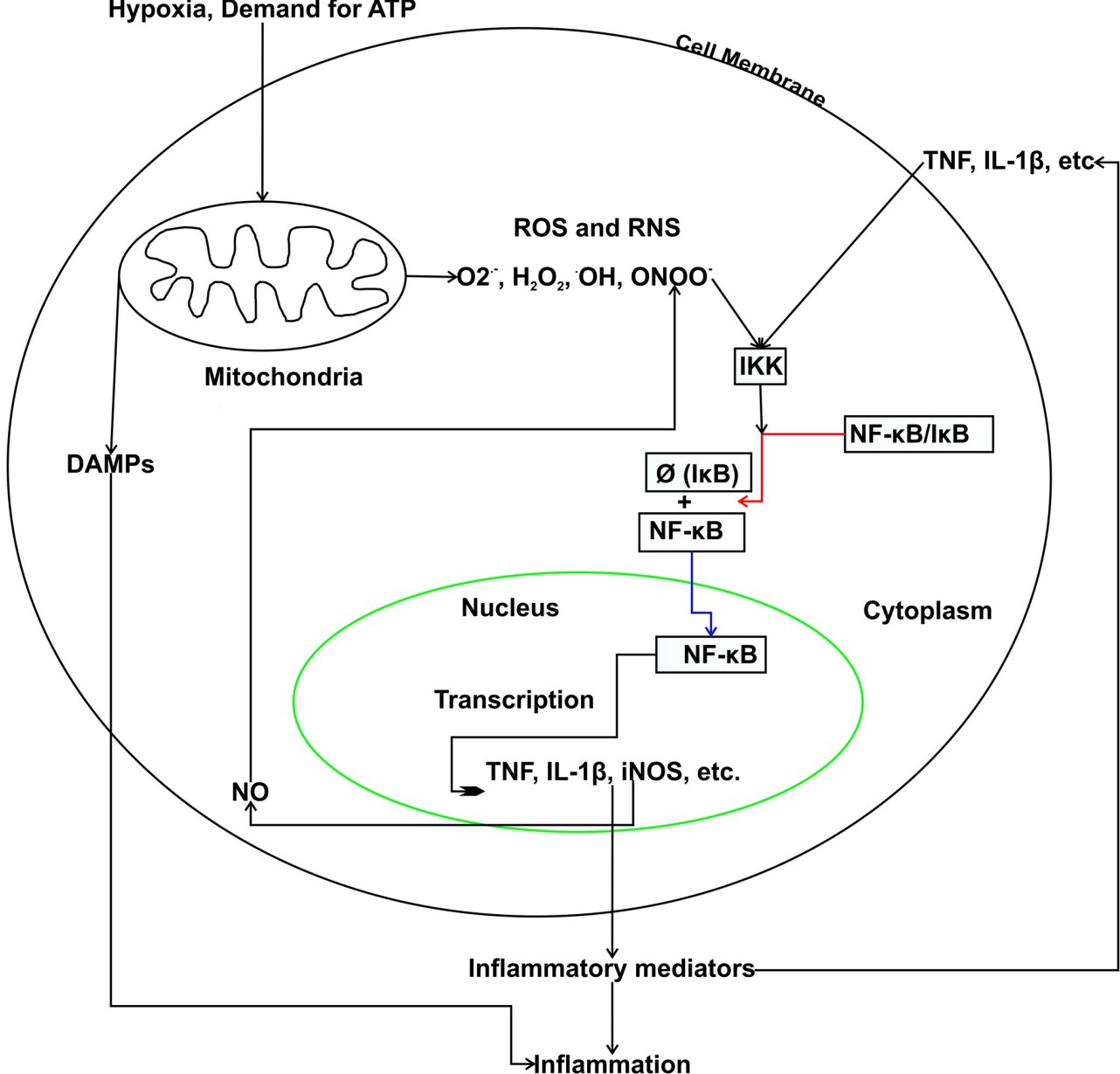

**Fig 5. The proposed model for the relation between mitochondrial dysfunction and inflammation in RA.** Hypoxia and demand for more ATP increase the production of mtROS and RNS, which activate the IKK enzyme that degrades IκB (degradation is represented with Ø). This results in the activation of transcription factor NF-κB that induces the expression of inflammatory mediators, such as tumour necrosis factor (TNF), interleukin-1 beta (IL-1β) and inducible nitric oxide synthase (iNOS). Further, the damage-associated molecular patterns (DAMPs) may also contribute to inflammation.

calcium-binding protein and known to promote cell death, was down-regulated [92]. The enzyme ACOT7, which increases the concentration of free fatty acids (FFA) by hydrolysing the acyl-CoA thioester of long-chain fatty acids, such as palmitoyl-CoA, was up-regulated in five datasets. This enzyme could be implicated in the remodelling of membrane phospholipids [93]. UCP2 uncouples OxPhos pathway and has been identified a candidate risk gene for RA by a whole genome association study [94]. This protein, which gets activated by $H_2O_2$ and $O_2^{\cdot-}$, was up-regulated in the microarray data. The uncoupling of OxPhos by UCP2 leads to

the dissipation of mitochondrial membrane potential gradient. Moreover, FFAs produced by ACOT7 are likely to play a role in the uncoupling by increasing the production of ROS [95].

A high ratio of kynurenine/tryptophan, which was observed in sera of RA patients, is needed for the kynurenine pathway for its role in anti-inflammation [96–97]. In our analysis, since KMO, which catalyses the hydroxylation of kynurenine to 3-hydroxykynurenine, was up-regulated in five datasets; it may deplete the concentration of kynurenine thereby impairing its activity in controlling inflammation. Further, this might enhance the generation of free radicals [57]. The analysis also identified the disruption of OxPhos: 11 subunits of OxPhos complexes were DEGs, of which seven were exclusively down-regulated, clearly indicating the negative regulation of OxPhos. This may further be involved in ROS production. Collectively, these results suggest that free radicals, negative regulation of OxPhos, and metabolic processes are highly active in RA synovial mitochondria.

Increased demand for ATP production on the respiratory machinery, and NOX enzymes are usually involved in ROS generation. But, the down-regulation of NOXs provides a strong support of ROS production by mitochondria rather than by the enzymes in the RA synovium. The observed down-regulation of the detoxifiers of ROS, such as CAT, GPX4 and SOD1 is consistent with other RA studies [98–100]. Since there is no up-regulation of any of these enzymes in any of the six microarray datasets, the detoxification of free radicals might be impaired. Further, LAP3, which degrades glutathione and plays a crucial role in cartilage and bone erosion, was up-regulated [101]. Apart from this, CASP8, an apoptotic caspase and a risk gene for RA, was up-regulated. But, in RA FLS, it is known to induce the activation of pro-inflammatory NF-κB and AP-1 transcription factors rather than apoptosis [45]. However, apoptosis may happen by a ROS-mediated increase in intracellular $Ca^{2+}$ levels, preferably through the SYK, BTK, BLNK and PLCγ2-mediated pathway. But it may nevertheless be insufficient to limit synovial hyperplasia. Thus, our results bring together enhanced oxidative stress and intrinsic apoptosis, with the up-regulation of processes involved in the generation of free radicals and with their impaired detoxification. These can be envisaged as a potential therapeutic strategy for RA. With regard to the network analysis, we show that UQCR10, MRPL4 and NDUFV2 are the three top hubs of the PPI network. UQCR10 belongs to the complex III, which is the middle segment of OxPhos, and is connected to 50 neighbours, of which nine were DEGs. This suggests a key role for this gene in the OxPhos pathway in RA synoviocytes. MRPL4, a part of the large 39S subunit of the mitochondrial ribosome, forms 49 PPI with neighbours. The complex I protein NDUFV2 is connected to 49 neighbours, including eight DEGs. Another complex I protein NDUFS6 has the maximum number of neighbour DEGs in the network. Since these proteins are the top hubs and have neighbour DEGs, it would be interesting to elucidate their roles in RA.

In this study, using publicly available microarray data, we discussed the roles of nDNA encoded proteins in RA mitochondrial dysfunction. Although the literature provides support to some of the inferences we made in the study, they need to be validated using experiments on cell lines or laboratory animals or in clinical studies. The expression levels of mtDNA encoded genes that were not probed by the microarray platforms could not be assessed in this study. Further, as the composition of cell types is different in both the healthy and RA synovial tissues, the changes in gene expression between them may be a manifestation of the respective cell types present in them.

## Conclusions

In conclusion, our study maximises the use of PPI and microarray data for studying mitochondrial dysfunction in RA. Identifying a set of nDNA encoded mitochondrial proteins implicated

in the dysregulation of pathways and processes associated with this organelle in the RA syno-vium was the idea behind this study. Analysing microarray data for identifying DEGs and understanding their likely functions in synovial mitochondria is highly informative in this context. Using DEGs, we identified the processes pertaining to the generation of free radicals and their impaired detoxification. The study also reports the possible occurrence of the mito-chondrion-mediated intrinsic apoptotic pathway in RA. We also, in particular, highlighted the roles of DEGs in the remodelling of membrane lipids, uncoupling electron transport and ATP synthesis, and amino acid and nucleotide metabolism in RA. We also proposed a model that links mitochondrial dysfunction to inflammation in RA by collating information from the lit-erature. These insights suggest several new routes for research into the role of mitochondria in RA. Particularly, oxidative stress and intrinsic apoptotic pathways may become attractive can-didates for new therapeutic interventions. However, our strategy herein was to develop a proof-of-principle method for studying mitochondrial dysfunction by integrating gene expres-sion, PPI, gene ontology and network analysis. Even though literature search has provided the possible implications for the study findings in RA mitochondrial dysfunction, their additional validation in experimental settings is needed.

## Supporting information

**S1 File. File for visualising the network using the Cytoscape tool.** The up- and down-regu-lated genes are highlighted in green and red colours respectively.
(XGMML)

**S1 Fig. A Venn diagram showing the translocation of the mitochondrial network genes to different parts of mitochondria.**
(TIF)

**S2 Fig. Hierarchical clustering of RA and control samples based on the gene expression of selected DEGs (Table 2) in GSE7307.**
(TIF)

**S3 Fig. Hierarchical clustering of RA and control samples based on the gene expression of selected DEGs (Table 2) in GSE55235.**
(TIF)

**S4 Fig. Hierarchical clustering of RA and control samples based on the gene expression of selected DEGs (Table 2) in GSE55457.**
(TIF)

**S5 Fig. Hierarchical clustering of RA and control samples based on the gene expression of selected DEGs (Table 2) in GSE12021 (HGU133A).**
(TIF)

**S6 Fig. Hierarchical clustering of RA and control samples based on the gene expression of selected DEGs (Table 2) in GSE12021 (HGU133B).**
(TIF)

**S7 Fig. Hierarchical clustering of RA and control samples based on the gene expression of selected DEGs (Table 2) in GSE77298.**
(TIF)

**S8 Fig. Hierarchical clustering of RA and control samples based on the gene expression of selected DEGs (Table 2) in GSE12021 (HGU133A) after removing RA samples clustered**

with control samples.
(TIF)

**S9 Fig. Hierarchical clustering of RA and control samples based on the gene expression of selected DEGs (Table 2) in GSE12021 (HGU133B) after removing RA samples clustered with control samples.**
(TIF)

**S10 Fig. Hierarchical clustering of prednisone treated and control samples (whole blood) based on the gene expression of selected DEGs (Table 2) normalized across samples in GSE77344.**
(TIFF)

**S11 Fig. Hierarchical clustering of celecoxib treated and control samples (colorectal primary adenocarcinomas)based on the gene expression of selected DEGs (Table 2) normalized across samples in GSE11237.**
(TIFF)

**S12 Fig. Hierarchical clustering of pre and post methotrexate treatment samples (early RA synovial biopsy) based on the gene expression of selected DEGs (Table 2) normalized across samples in GSE45867.**
(TIFF)

**S13 Fig. The distribution of the number of all neighbours and DEG neighbours of all the proteins of the mitochondrial PPI network.**
(TIF)

**S14 Fig. The subnetwork of UQCR10.**
(TIF)

**S15 Fig. The subnetwork of NDUFV2.**
(TIF)

**S16 Fig. The subnetwork of NDUFS6.**
(TIF)

**S17 Fig. The subnetwork of UQCRQ.**
(TIF)

**S1 Table. The number of microarray datasets in which the mitochondrial PPI network genes were differentially expressed.**
(XLS)

**S2 Table. The number of microarray datasets in which, the MitoCarta genes, which are not part of the network, were differentially expressed.**
(XLSX)

**S3 Table. Number of neighbours for all mitochondrial PPI network proteins.** The table shows the number of first neighbours, the number of DEGs and number of up/down regulated DEGs among the first neighbours.
(XLSX)

**S4 Table. The significantly enriched biological processes (BP) of the RA synovial mitochondrial DEGs.**
(XLS)

## Acknowledgments

We thank the Institute of Bioinformatics and Applied Biotechnology (IBAB) for providing facilities and the environment for carrying out this research work.

## Author Contributions

**Conceptualization:** Venugopal Panga, Srivatsan Raghunathan.

**Formal analysis:** Venugopal Panga, Ashwin Adrian Kallor, Arunima Nair, Shilpa Harshan, Srivatsan Raghunathan.

**Investigation:** Venugopal Panga, Ashwin Adrian Kallor, Arunima Nair, Shilpa Harshan, Srivatsan Raghunathan.

**Methodology:** Venugopal Panga, Ashwin Adrian Kallor, Arunima Nair, Srivatsan Raghunathan.

**Project administration:** Srivatsan Raghunathan.

**Resources:** Srivatsan Raghunathan.

**Software:** Venugopal Panga, Ashwin Adrian Kallor, Arunima Nair, Shilpa Harshan.

**Supervision:** Srivatsan Raghunathan.

**Validation:** Shilpa Harshan.

**Visualization:** Venugopal Panga, Shilpa Harshan.

**Writing – original draft:** Venugopal Panga.

**Writing – review & editing:** Shilpa Harshan, Srivatsan Raghunathan.

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
