## [Decision Letter · Decision Letter 0]

6 Aug 2019

PONE-D-19-14453

Mitochondrial Dysfunction in Rheumatoid Arthritis: A Comprehensive Analysis by Integrating Gene Expression, Protein-Protein Interactions and Gene Ontology Data

PLOS ONE

Dear Dr. Srivatsan Raghunathan,

Thank you for submitting your manuscript to PLOS ONE. After careful consideration, we feel that it has merit but does not fully meet PLOS ONE’s publication criteria as it currently stands. Therefore, we invite you to submit a revised version of the manuscript that addresses the points raised during the review process.

Both referees appreciated the amount of work and the results. You will see that reviewer #2 asked to better clarify the usage of databases and their limitations, and asked for textual changes and clarifications that will reflect what this manuscript adds as compared tom previous work in the field. Referee #1 suggested some experiments, of which the first is important for publication of the current manuscript. I ask you to carefully revise your manuscript while replying in a point-by-point manner to each of the comments raised. 

We would appreciate receiving your revised manuscript by Sep 20 2019 11:59PM. To enhance the reproducibility of your results, we recommend that if applicable you deposit your laboratory protocols in protocols.io, where a protocol can be assigned its own identifier (DOI) such that it can be cited independently in the future. For instructions see: http://journals.plos.org/plosone/s/submission-guidelines#loc-laboratory-protocols

We look forward to receiving your revised manuscript.

Kind regards,

Dan Mishmar

Academic Editor

PLOS ONE

Journal Requirements:

 [We thank Department of IT, BT and S&T, Government of Karnataka, India for infrastructure support. VP received fellowship from IBAB as well as from the council of scientific and industrial research (CSIR), GoI (File No: 09/1086(0001)/2012-EMR-I), URL: http://www.csirhrdg.res.in/. SR is a faculty at IBAB and this project was partially supported by a grant from the Department of Biotechnology, GoI (BTPR12422/MED/31/287/2014), URL: http://www.dbtindia.nic.in/.]. 

Reviewers' comments:

Reviewer's Responses to Questions

**Comments to the Author**

1. Is the manuscript technically sound, and do the data support the conclusions?

Reviewer #1: Partly

Reviewer #2: Yes

2. Has the statistical analysis been performed appropriately and rigorously? 

Reviewer #1: Yes

Reviewer #2: Yes

3. Have the authors made all data underlying the findings in their manuscript fully available?

Reviewer #1: Yes

Reviewer #2: Yes

4. Is the manuscript presented in an intelligible fashion and written in standard English?

Reviewer #1: Yes

Reviewer #2: Yes

5. Review Comments to the Author

Reviewer #1: The manuscript by Panga and colleagues assesses the mitochondrial dysfunction in Rheumatoid Arthritis (RA). This is an interesting and evolving field exploring the key role of mitochondrial dysfunction in inflammatory disease such as RA. The authors performed a set of integrative analyses of gene expression, protein-protein interactions, and gene ontology from existing data and literature. The data suggested an additional role of nDNA encoded proteins in mitochondrial dysfunction and their relation to inflammation in RA. The authors provided an elegant, important, and very useful tool to analyze nDNA encoded proteins that related to mitochondrial dysfunction in specific disease (in this case, RA).

Major point:

The main concern in this study is that the treatment options for RA as mentioned in Table 3 are known to affect mitochondrial function. Therefore, the question becomes how can the authors distinguish between the effects of the disease versus the treatments? The following are two suggested additional experiments that may help further elucidate this concern: (1) test the expression of the same candidate genes in different tissues from patients with non-RA-related diseases that receive the same treatment (if available). (2) test the various treatment effects on mitochondrial function and the expression of the candidate genes (listed on table 2) in an in vitro model such 293T cell lines.

Minor points:

Reviewer #2: PLoS One, Rheumatoid Arthritis

Several studies have reported mitochondrial dysfunction in rheumatoid arthritis (RA). A number of these have been rather poorly supported haplogroup association studies. This paper has taken a different (wider) approach with an analysis of gene expression, protein-protein interactions (PPI) and gene ontology data to consider the role of mitochondria in RA. I am a supporter of moving away from narrow haplogroup approaches in the context of the investigation of mitochondrial dysfunction in complex traits. However, I have not worked on this phenotype in the past.

This paper presents a substantial body of data.

The limitations of the datasets used should be clearly discussed.

“In this study, we created, for the first time, a PPI network that is specific to RA synovial mitochondria” I have noted there are a number of papers reporting PPI networks for RA. Could the authors be more specific as to how the work here differs from the prior work, or builds upon it?

“We also hypothesised a process by which mitochondrial dysfunction could lead to inflammation in RA by collating information from the literature” So, this is not a novel hypothesis, give the central references used in the formulation of this hypothesis.

“However, our strategy herein was to develop a proof-of-principle method for studying mitochondrial dysfunction by integrating gene expression, PPI, gene ontology and network analysis”Similar approaches have been applied by a number of other groups in the past such as the Mootha lab for a number of years. It would be appropriate to mention the work of this group, in addition to citation 6. I note that you have also published similar work recently “A cytokine protein-protein interaction network for identifying key molecules in rheumatoid arthritis. Panga V.” This used the same datasets, correct? I assume the method is similar but with the focus here being the mitochondria rather than cytokines.

Other points

Table 3 – this data can be shown in a more compact format

Table 9 - really not required

Overall this data is likely to be of interest to those studying RA. I think the paper can place the work in the context of prior work more clearly. The length can also be reduced by working on the format of some of the tables.

6. PLOS authors have the option to publish the peer review history of their article (what does this mean?). If published, this will include your full peer review and any attached files.

Reviewer #1: No

Reviewer #2: No

---

## [Author Response · Author response to Decision Letter 0]

20 Sep 2019

Reviewer #1: The manuscript by Panga and colleagues assesses the mitochondrial dysfunction in Rheumatoid Arthritis (RA). This is an interesting and evolving field exploring the key role of mitochondrial dysfunction in inflammatory disease such as RA. The authors performed a set of integrative analyses of gene expression, protein-protein interactions, and gene ontology from existing data and literature. The data suggested an additional role of nDNA encoded proteins in mitochondrial dysfunction and their relation to inflammation in RA. The authors provided an elegant, important, and very useful tool to analyze nDNA encoded proteins that related to mitochondrial dysfunction in specific disease (in this case, RA).

Major point:

The main concern in this study is that the treatment options for RA as mentioned in Table 3 are known to affect mitochondrial function [1-3]. Therefore, the question becomes how can the authors distinguish between the effects of the disease versus the treatments? The following are two suggested additional experiments that may help further elucidate this concern:

(1) test the expression of the same candidate genes in different tissues from patients with non-RA-related diseases that receive the same treatment (if available).

(2) test the various treatment effects on mitochondrial function and the expression of the candidate genes (listed on table 2) in an in vitro model such 293T cell lines.

Our response:

(1) According to the references [1-3] provided by the reviewer, the drugs such as Sulfasalazine (Azulfidine), Prednisolone and Methotrexate (MTX) affect mitochondrial function. As per the first reference [1], Sulfasalazine induces mitochondrial dysfunction in rat kidneys. According to reference [2], Prednisolone enhances the production of mitochondrial ROS (mtROS) in human CEC cell line (hCECs). Reference [3] says that MTX causes mitochondrial injury in rat hepatocytes. In the microarray datasets used in this analysis, combinations of these three drugs along with others were used for treating RA patients (Table 3 of the manuscript).

1. Following the suggestion of the reviewer, we first searched for a gene expression experiment where the same combination of drugs was used to treat non-RA related diseases. Unfortunately we could not come across any such experiment. As a next option, we searched for experiments where individual drugs were used for treating non-RA diseases and could come across the following experiments:

 (i) Whole blood gene expression data from COPD patients, treated and not treated with prednisone, which is a prodrug for prednisolone (GEO microarray data set GSE77344) [4]. 

 (ii) colorectal primary adenocarcinomas surgically removed from 23 patients, 11 of whom received 400 mg celecoxib two times per day for 7 days prior to surgery and 12 who did not receive the treatment. Celecoxib is a COX-2 inhibitor used to treat RA (GEO microarray data set GSE11237) [5].

 (iii) In addition, we came across the dataset that had paired synovial tissue biopsies from early RA patients naive to therapy, pre and 12 weeks post initiation of methotrexate therapy (GEO microarray data set GSE45867) [6]. This has direct relevance to our discussion on the effect of drug versus disease on candidate gene expression.

We could not find such experiments involving the drug Azulfidine.

We analysed the differential expression in these datasets and checked the status of the candidate genes using the same criteria as the original RA datasets (fold change < |1.5|, pvalue >= 0.05). The heatmaps of normalized expression values for the candidate genes is shown in figures S10-S12 in the revised manuscript.

In GSE77344, which compared prednisone treated and untreated COPD patients, a single gene, MAOA showed considerable upregulation (fold change = 3.9, pvalue = 0.001) by RMA normalization. (MAS5 normalization could not performed on this dataset because it uses the platform Human Gene 1.1 ST Array [transcript (gene) version], which does not have mismatch probes). No candidate genes listed in Table 2 of the manuscript were differentially regulated in GSE11237 or GSE45867. The hierarchical clustering of samples based on the expression values of genes normalized across samples also do not show a separation of treated from untreated, or pre treatment from post treatment (Figures S10-S12).

Based on these results we conclude that there was negligible effect on the candidate genes when the patients were treated individually with prednisone, methotrexate or celecoxib. 

In one of the dataset (GSE7307) used in the current study, RA patients were not treated with drugs but still the differential expression of the candidate genes listed in Table 2 was observed (Fig S2) .

In our study we have shown that the expression of the candidate genes is affected by the combined drug treatment in two datasets (GSE12021 (HGU133A) and GSE12021 (HGU133B)). In these two datasets, few RA samples which were treated with drugs have clustered with the healthy controls, showing that there is a drug effect (lines 314-332 of the revised manuscript). Therefore, we have removed the RA samples clustered with healthy controls before testing the effects of these drugs on gene expression in the other RA samples. After removing the RA samples clustered with healthy controls in the two datasets, we observed the complete separation of healthy and RA samples in the clusters (S8 and S9 Figs). We hope this step takes care of any combinatorial drug effects.

We have edited the manuscript to include the above arguments [lines 333-351].

(2) We thank the reviewer for this important comment. However, we feel that testing various treatment effects on mitochondrial function and the expression of the candidate genes (Table 2 of the manuscript) in an in vitro model such as 293T cell lines is beyond the scope of the present project. We will certainly consider this suggestion in our future studies.

Reviewer #1

Minor points:

 1. Somatic mutations in the mtDNA were found in RA patients [7]. Testing for somatic mutations in the mtDNA and assessment of the expression level of mtDNA encoded genes may provide additional support to your hypothesis.

 2. “We also hypothesized a process by which mitochondrial dysfunction could lead to inflammation in RA” is not a new hypothesis [8].

Our response:

 (1) We thank the reviewer for this valuable comment. It is widely known that all the protein-coding genes of the mtDNA (total 13) are involved in the OxPhos pathway. As stated in lines 446-447 of the revised manuscript:

Lines 446-447: “The mitochondrial protein-coding genes of OxPhos were either missed or non-DEGs in the microarray studies.” 

The genes were missed because the microarray platforms did not have probes to measure them. Therefore, we could not assess the expression levels of these genes. Since we only use microarray data in this analysis, we could not study mtDNA mutations.

However, in future, we look forward to testing for somatic mutations in the mtDNA genes and assessing their expression levels using sequencing technology or specialised microarray techniques.

 (2) Yes, this is not a new hypothesis. Sorry for the wording. Actually, this refers to the proposed model that explains the link between mitochondrial dysfunction and inflammation during RA (line 478 of the revised manuscript, subheading : “A model relating mtROS and inflammation in the RA synovium”). In the model, we say that the production of ROS and their involvement in NF-κB activation, increased glutathione degradation (as evidenced by the consistent up-regulation of LAP3 gene in five microarray datasets) and DAMPs released from mitochondria have connections to inflammation.

The references [43, 76-82] in the manuscript are in support of the model. Therefore, to avoid confusion, in the revised manuscript, we modified the sentence as follows:

Lines 571-573: “We also proposed a model that links mitochondrial dysfunction to inflammation in RA by collating information from the literature. “

Reviewer #2: Several studies have reported mitochondrial dysfunction in rheumatoid arthritis (RA). A number of these have been rather poorly supported haplogroup association studies. This paper has taken a different (wider) approach with an analysis of gene expression, protein-protein interactions (PPI) and gene ontology data to consider the role of mitochondria in RA. I am a supporter of moving away from narrow haplogroup approaches in the context of the investigation of mitochondrial dysfunction in complex traits. However, I have not worked on this phenotype in the past. This paper presents a substantial body of data.

The limitations of the datasets should be clearly discussed.

Our response:

Thank you for your appreciation. In this study, we used microarray datasets. Here, we discuss some of the limitations of these datasets.

 1. Microarray, which is a high-throughput gene expression profiling technique, offers a great predictive power for understanding the roles of gene expression in diseases. In this study, using publicly available microarray data, we discussed the roles of nDNA encoded proteins in RA mitochondrial dysfunction. Although the literature provides support to some of the inferences we made in the study, they need to be validated using experiments on cell lines or laboratory animals or in clinical studies. Further, the microarray platforms used in the datasets did not include probes for mtDNA. Hence no inference could be made about their role in RA inflammation.

 2. The composition of cell types in a healthy synovium is different to that of RA. The healthy synovium primarily contains two cell types, macrophage-like synoviocytes (MLS) and fibroblast-like synoviocytes (FLS) [9]. Other cell types such as leucocytes can be seen in small numbers [9]. In contrast, the RA synovium is expanded and forms pannus and contains resident MLS and FLS as well as heavily infiltrated leucocytes [10-11]. Therefore, the changes in gene expression between the two synovial samples maybe a manifestation of the respective cell types present in them.

Now, the composition of cell types in the healthy and RA synovial tissues is included in the ‘Introduction’ section between the lines 62 and 66 of the revised manuscript. 

Lines 62-66: “The composition of cell types in a healthy synovium is different to that of RA. The healthy synovium primarily contains two cell types, macrophage-like synoviocytes (MLS) and fibroblast-like synoviocytes (FLS) [19]. Other cell types such as leucocytes can be seen in small numbers [19]. In contrast, the RA synovium is expanded and forms pannus and contains resident MLS and FLS as well as heavily infiltrated leucocytes [20-21].”

The limitations are discussed in the ‘Discussion’ section between the lines 553 and 560 of the revised manuscript.

Lines 553-560: “In this study, using publicly available microarray data, we discussed the roles of nDNA encoded proteins in RA mitochondrial dysfunction. Although the literature provides support to some of the inferences we made in the study, they need to be validated using experiments on cell lines or laboratory animals or in clinical studies. The expression levels of mtDNA encoded genes that were not probed by the microarray platforms could not be assessed in this study. Further, as the composition of cell types is different in both the healthy and RA synovial tissues, the changes in gene expression between them may be a manifestation of the respective cell types present in them.”

Reviewer #2: “In this study, we created, for the first time, a PPI network that is specific to RA synovial mitochondria” I have noted there are a number of papers reporting PPI networks for RA. Could the authors be more specific as to how the work here differs from the prior work, or builds upon it?

Our response:

Many researchers had created PPI networks for RA at the level of a cell. For instance, in order to identify key molecules, earlier we had created a PPI network for cytokine signalling in RA [12]. Similarly, few other studies had identified highly connected regions, ego networks and key genes in PPI networks in RA and other diseases [13-15]. Another study used PPI for determining the efficacy of leflunomide and ligustrazine drugs in the treatment of RA [16]. In contrast, in this study, we created, for the first time, a PPI network that is specific to RA synovial mitochondria and have identified hub proteins. Therefore, this network is at the level of a cell organelle (mitochondrion) while other networks are at the level of a cell. Now, these details are updated in the discussion section of the revised manuscript between the lines 501 and 505.

Lines 501-505: “Many researchers had created PPI networks for RA at the level of a cell. For instance, in order to identify key molecules, earlier we had created a PPI network for cytokine signalling in RA [83]. Similarly, few studies had identified highly connected regions, ego networks and key genes in PPI networks in RA and other diseases [84-86]. Another study used PPI for determining the efficacy of leflunomide and ligustrazine drugs in the treatment of RA [87]. ”

Reviewer #2: “We also hypothesised a process by which mitochondrial dysfunction could lead to inflammation in RA by collating information from the literature” So, this is not a novel hypothesis, give the central references used in the formulation of this hypothesis.

Our response:

Yes, this is not a novel hypothesis. Sorry for the wording. Actually, this refers to the proposed model that explains the link between mitochondrial dysfunction and inflammation in RA (line 478 of the revised manuscript). In the model, we say that the production of ROS and their involvement in NF-κB activation, increased glutathione degradation (as evidenced by the consistent up-regulation of LAP3 gene in five microarray datasets) and DAMPs released from mitochondria have connections to inflammation.

The references, 43, 76-82, which are cited between the lines 479 and 492 in the revised manuscript, are in support of the model. Therefore, to avoid confusion, in the revised manuscript, we modified the sentence to “we also proposed a model that links mitochondrial dysfunction to inflammation in RA by collating information from the literature” (lines 571-573).

Reviewer #2: “However, our strategy herein was to develop a proof-of-principle method for studying mitochondrial dysfunction by integrating gene expression, PPI, gene ontology and network analysis” Similar approaches have been applied by a number of other groups in the past such as the Mootha lab for a number of years. It would be appropriate to mention the work of this group, in addition to citation 6. I note that you have also published similar work recently “A cytokine protein-protein interaction network for identifying key molecules in rheumatoid arthritis. Panga V.” This used the same datasets, right? I assume the method is similar but with the focus here being the mitochondria rather than cytokines.

Our response:

The reviewer is quite right about our earlier paper for which we used the same datasets. Yes, the method is similar but with focus here being the mitochondrial dysfunction rather than cytokines.

Several aspects of mitochondrial biology both in health and disease have been uncovered by the Mootha lab over a number of years. The compendium of mitochondrial proteins (MitoCarta) was one of the major contributions from their lab (reference 16 of the revised manuscript). Their group, using genomic technologies and computational approaches, has studied mitochondrial dysfunction in several diseases such as Leigh syndrome, cardiovascular diseases, obesity and infantile-onset mitochondrial encephalopathy [17-23]. They identified metabolic abnormalities concerned with mitochondria and biallelic mutations leading to instability in mitoribosomal subunits in Leigh syndrome [17, 19]. Their works on mitochondrial calcium uniporter have greater importance both in mitochondrial biology and several associated diseases (for instance in diastolic heart disease) [18, 24-26]. Now, we included these details in the ‘Introduction’ section of the revised manuscript (lines 35-43).

Lines 35-43; “Genomic technologies and computational approaches played a vital role in our understanding of mitochondrial dysfunction in several diseases like Leigh syndrome, cardiovascular diseases, obesity and infantile-onset mitochondrial encephalopathy [6-12]. These approaches have also discerned the mechanics of calcium uniporter in mitochondrial biology and associated diseases [7, 13-15]. Further, investigations into metabolic profiling and whole-exome sequencing data point to metabolic abnormalities concerned with mitochondria and biallelic mutations leading to instability in mitoribosomal subunits in Leigh syndrome [6, 8].”

Reviewer #2: Other points

Table 3 – this data can be shown in a more compact format.

Table 9 – really not required

Overall this data is likely to be of interest to those studying RA. I think the paper can place the work in the context of prior work more clearly. The length can also be reduced by working on the format of some of the tables.

Our response:

We thank the reviewer for this comment. As per the suggestion, in the revised manuscript, we have shown Table 3 in a compact format and Table 9 (of the original manuscript) has been removed. We also reduced the length of the manuscript by working on the tables.

---

## [Decision Letter · Decision Letter 1]

18 Oct 2019

Mitochondrial dysfunction in rheumatoid arthritis: A comprehensive analysis by integrating gene expression, protein-protein interactions and gene ontology data

PONE-D-19-14453R1

Dear Dr. Raghunathan,

We are pleased to inform you that your manuscript has been judged scientifically suitable for publication and will be formally accepted for publication once it complies with all outstanding technical requirements.

With kind regards,

Dan Mishmar

Academic Editor

PLOS ONE

Additional Editor Comments (optional):

Reviewers' comments:

Reviewer's Responses to Questions

**Comments to the Author**

1. If the authors have adequately addressed your comments raised in a previous round of review and you feel that this manuscript is now acceptable for publication, you may indicate that here to bypass the “Comments to the Author” section, enter your conflict of interest statement in the “Confidential to Editor” section, and submit your "Accept" recommendation.

Reviewer #1: All comments have been addressed

Reviewer #2: All comments have been addressed

2. Is the manuscript technically sound, and do the data support the conclusions?

Reviewer #1: Yes

Reviewer #2: Yes

3. Has the statistical analysis been performed appropriately and rigorously? 

Reviewer #1: Yes

Reviewer #2: Yes

4. Have the authors made all data underlying the findings in their manuscript fully available?

Reviewer #1: Yes

Reviewer #2: Yes

5. Is the manuscript presented in an intelligible fashion and written in standard English?

Reviewer #1: Yes

Reviewer #2: Yes

6. Review Comments to the Author

Reviewer #1: (No Response)

Reviewer #2: I have looked at the amended paper and my comments have been addressed the article should proceed to publication.

7. PLOS authors have the option to publish the peer review history of their article (what does this mean?). If published, this will include your full peer review and any attached files.

Reviewer #1: No

Reviewer #2: No

---

## [Editor Report · Acceptance letter]

28 Oct 2019

PONE-D-19-14453R1 

Mitochondrial dysfunction in rheumatoid arthritis: A comprehensive analysis by integrating gene expression, protein-protein interactions and gene ontology data 

Dear Dr. Raghunathan:

I am pleased to inform you that your manuscript has been deemed suitable for publication in PLOS ONE. Congratulations! Your manuscript is now with our production department. 

With kind regards,

on behalf of

Dr. Dan Mishmar 

Academic Editor

PLOS ONE